# Silicone Implants Immobilized with Interleukin-4 Promote the M2 Polarization of Macrophages and Inhibit the Formation of Fibrous Capsules

**DOI:** 10.3390/polym13162630

**Published:** 2021-08-07

**Authors:** Hyun-Seok Kim, Seongsoo Kim, Byung-Ho Shin, Chan-Yeong Heo, Omar Faruq, Le Thi Van Anh, Nilsu Dönmez, Pham Ngoc Chien, Dong-Sik Shin, Sun-Young Nam, Rong-Min Baek

**Affiliations:** 1Department of Plastic and Reconstructive Surgery, Seoul National University Hospital, Seoul National University College of Medicine, Seoul 03080, Korea; hsmercy@naver.com (H.-S.K.); lionheo@gmail.com (C.-Y.H.); 2Division of Chemical and Bioengineering, Kangwon National University, Chuncheon 24341, Korea; ksswdf@gmail.com; 3Department of Plastic and Reconstructive Surgery, Seoul National University Bundang Hospital, Seongnam 13620, Korea; shinzmatt@naver.com (B.-H.S.); ofaruq1991@gmail.com (O.F.); vananhle2201@gmail.com (L.T.V.A.); nlsdonmez@gmail.com (N.D.); ngocchien1781@gmail.com (P.N.C.); 4Department of Chemical and Biological Engineering, Sookmyung Women’s University, Seoul 04310, Korea; 5Industry Collaboration Center, Sookmyung Women’s University, Seoul 04310, Korea

**Keywords:** silicone implants, immobilization of IL-4, capsular contracture, fibrosis, inflammation, macrophage polarization

## Abstract

Breast augmentations with silicone implants can have adverse effects on tissues that, in turn, lead to capsular contracture (CC). One of the potential ways of overcoming CC is to control the implant/host interaction using immunomodulatory agents. Recently, a high ratio of anti-inflammatory (M2) macrophages to pro-inflammatory (M1) macrophages has been reported to be an effective tissue regeneration approach at the implant site. In this study, a biofunctionalized implant was coated with interleukin (IL)-4 to inhibit an adverse immune reaction and promoted tissue regeneration by promoting polarization of macrophages into the M2 pro-healing phenotype in the long term. Surface wettability, nitrogen content, and atomic force microscopy data clearly showed the successful immobilization of IL-4 on the silicone implant. Furthermore, in vitro results revealed that IL-4-coated implants were able to decrease the secretion of inflammatory cytokines (IL-6 and tumor necrosis factor-α) and induced the production of IL-10 and the upregulation of arginase-1 (mannose receptor expressed by M2 macrophage). The efficacy of this immunomodulatory implant was further demonstrated in an in vivo rat model. The animal study showed that the presence of IL-4 diminished the capsule thickness, the amount of collagen, tissue inflammation, and the infiltration of fibroblasts and myofibroblasts. These results suggest that macrophage phenotype modulation can effectively reduce inflammation and fibrous CC on a silicone implant conjugated with IL-4.

## 1. Introduction

The 2017 Plastic Surgery Statistics Report announced that there were 300,378 breast implant operations performed that year, showing an increase of 3% compared to the year 2016 [1]. Currently, silicone breast implants are the most common and approved means of breast augmentation [2]. However, the medical literature reports that silicone breast implants are linked to severe adverse effects on health [3]. Capsular contracture (CC) is one of the primary emerging complications of alloplastic breast reconstruction. Thus far, a 10.6% incidence of CC has been associated with silicone breast implants [4]. CC is a multifactorial fibrotic response that results in an increase in capsule stiffness for the connection between the tissue and the implant, causing discomfort and aching after augmentation mammoplasty [5]. The specific cellular components of implant capsules, such as macrophages, fibroblasts, and lymphocytes, seem to influence the development of the fibrous capsules. However, the disease of CC has not been fully explained [3,6].

Macrophages are phenotypically diverse and quite abundant immune cell populations present at defect sites during tissue regeneration and remodeling processes [7]. They arrive at a site of injury within 24 h and reach a peak within 14–21 days [8]. After insertion of implants, macrophages are recruited from circulation around the implant and are responsible for the bulk of phagocytosis, debris removal, biomolecule production, and remodeling of the extracellular matrix (ECM) [9]. In recent years, it has been reported that macrophages are associated with a different spectrum of activation states/phenotypes, which has led to their broad categorization as pro-inflammatory (M1) or pro-healing (M2) macrophages [10]. Injury-triggered endogenous inflammatory signals such as those from T-helper cells and the cytokine IFN-γ activate M1 macrophages, which are actively involved in inflammation and tumor destruction [11]. Moreover, M1 macrophages produce high levels of pro-inflammatory cytokines and oxidative metabolites (e.g., nitric oxide and superoxide). In contrast, M2 macrophages are induced by Th2 cytokines, such as IL-13, IL-10, and IL-4, and profoundly support tissue repair and growth. They produce anti-inflammatory cytokines that participate in matrix remodeling, angiogenesis, and cell replacement [12]. While M1 macrophages facilitate and are required for biomaterial implantation, prolonged M1 macrophage exposure causes severe reactions to foreign bodies, granulomas, and fibrous encapsulation, leading to chronic inflammatory responses and a lack of assimilation of the biomaterial [13,14].

Some researchers have reported that a high ratio of M2 to M1 macrophages leads to effective tissue repair, absorption, and regeneration at the implant area [15]. In light of the important role of macrophage polarization in inflammation and capsular contracture around silicone implants, we propose the hypothesis that a local presence of IL-4—a Th2 cytokine inducing M2 macrophages—on silicone implants may protect against capsular contracture by skewing macrophages to the anti-inflammatory M2 phenotype. IL-4, a multifunctional pleiotropic cytokine, is expressed mainly by activated T cells but also by eosinophils, basophils, and mast cells. It maintains the typical cytokine structure (high sequence homology with other cytokines) and shares cell surface receptors and intracellular signaling with other cytokines. M2 macrophages are also induced by other cytokines, such as IL-10 and IL-13. However, IL-4 is crucial for CD4+ Th2 cell functionalities, whereas IL-13 has been demonstrated to be less effective on T cells [16,17]. Some other studies have reported that IL-4 is more effective than IL-13. On the other hand, IL-4 drastically enhanced IL-10 production [18]. IL-4 is best known for defining the Th2 phenotype and for maintaining apoptosis, cell proliferation, and the expression of myriad genes in numerous cell types, including fibroblasts, macrophages, lymphocytes, and endothelial cells [19].

To assess our hypothesis, silicone implant surface modification with IL-4 was performed. The surface modification steps, including O_2_ plasma treatment, functionalization of aminosilane (APTMS), and IL-4 conjugation, were serially characterized by surface wettability and nanoscale topography. In addition, we cultured and stimulated RAW 264.7 cells on silicone implants with or without IL-4 to search for possible pathways in vitro. We also conducted an animal study in which we inserted one silicone implant each into the subpanniculus planes of rats for a definite period of time. The tissue around the implant was extracted, and it was examined by hematoxylin and eosin (H&E) staining, Masson’s trichrome (MT) staining, and immunofluorescence (IF) staining.

## 2. Materials and Methods

### 2.1. Materials

Therapeutic silicone breast implants were kindly provided by Hans Biomed (Seoul, Korea). Polyvinyl alcohol (PVA, 98–99% hydrolyzed, MW = 31,000–50,000), poly ethylene glycol diamine (PEG diamine, MW = 2000), *N*-(3-dimethylaminopropyl)-*N*′-ethylcarbodiimide hydrochloride (EDC), *N*-hydroxysulfosuccinimide sodium salt (sulfo-NHS), sodium dodecyl sulphate (SDS), D-mannose, Tween 80, diclofenac sodium salt (DF), dimethyl sulfoxide (DMSO), thiazolyl blue tetrazolium bromide solution (MTT solution), sodium bicarbonate, bovine serum albumin (BSA), (3-aminopropyl)trimethoxysilane (APTMS), acetonitrile (ACN), *N*,*N*-diisopropylethylamine (DIPEA), and methanol were acquired from Sigma–Aldrich (Saint louis, MO, USA). EPO-TEK 301-2 medical epoxy was collected from Epoxy Technology (Billerica, MA, USA). Quanta Biodesign (Plain City, OH, USA) provided Bis-dPEG^®^ _13_-NHS ester. Dulbecco’s phosphate-buffered saline (DPBS) was obtained from Welgene (Gyeongsangbuk-do, Gyeongsan, Korea). For in vitro cell culture, DMEM/high glucose, fetal bovine serum (FBS), and penicillin/streptomycin were provided by Hyclone (Logan, UT, USA). Invitrogen (Carlsbad, CA, USA) contributed Hoechst 33,342 for nucleus staining. Anti-mouse IL-6, TNF-α, IL-4, and IL-10 antibodies (Abs); biotinylated anti-mouse IL-4, TNF-α, IL-10, and IL-6 Abs; recombinant mouse IL-6, TNF-α, IL-4, and IL-10 Abs; CD206; CD68; CD11b; Arg-1; and iNOS Ab were obtained from Abcam (Cambridge, MA, USA). Paraformaldehyde (4%) was received from KCFC (Seoul, Korea). For histological staining, xylene, ethanol, and hydrochloric acid (35–37%) were collected from Duksan Pure Chemicals (Ansan, Korea). Junsie supplied the ammonia solution (28–30%). Modified Mayer’s H&E Y solutions were purchased from Richard–Allan Scientific (Kalamazoo, MI, USA). For Masson’s trichrome (MT) staining, acetic acid (1%) and Biebrich scarlet-acid fuchsin were purchased from Duksan Pure Chemicals (Ansan, Korea), and other agents, including phosphomolybdic acid and aniline blue, were purchased from Sigma–Aldrich. The immunofluorescence staining solution (10×) was provided by Dako (Glostrup, Denmark). Abcam (Cambridge, MA, USA) supplied anti-vimentin (ab92547) and anti-α-SMA (ab5694). Paraffin was purchased from Merck (Kenilworth, NJ, USA). 

### 2.2. Immobilization of IL-4 on Silicone Implant

Before surface treatment, silicone implants were sterilized using ethylene oxide, and then the shell-type silicone implants (thickness: 1.5 mm; diameter: 2 cm) were rinsed with MeOH and dried in vacuo. Each silicone implant was treated in an oxygen plasma chamber (15 standard cubic centimeter per minute (sccm), 100 W, 15 min). The oxygen-plasma-treated silicone implant was then immersed in 0.5 (wt)% APTMS of acetonitrile solution and incubated (100 rpm) for 2 h at RT. ACN solution was used to wash the silicone implant three times, followed by drying under N_2_ gas flow. To a solution of 5 mM Bis-dPEG^®^ _13_-NHS ester and 10 mM DIPEA in ACN, the amino-functionalized silicone implant was added and incubated (100 rpm) for 2 h at RT. The silicone implant was then rinsed with ACN solution three times before drying under N2 gas flow. The NHS-functionalized silicone implant was added to bicarbonate buffer solution (100 mM, pH 8.2), including 0.5 mg/mL IL-4 and 1% (*w*/*v*) BSA and incubated (100 rpm) for 1 h at RT. The IL-4-immobilized silicone implant was then blow-dried using N_2_ gas after washing three times with PBS buffer (5 mM, pH 7.4). The entire process remained sterile.

### 2.3. The Physicochemical Characterization of Modified Silicone Implants

The nitrogen content present in the silicone implant surface was evaluated by energy dispersive X-ray spectroscopy (EDS) analysis. An electron source was irradiated on the silicone implant surface using a thermal Schottky field emitter with an electron beam resolution of 0.8 nm, 15 Kv. Every region used for calculating nitrogen content was measured in triplicate and averaged; the composition of the outermost layer of the silicone-terminated Si surface after salinization with APTMS was used.

For estimation of the surface wettability, the water contact angles (WCAs) of the modified silicone implants were measured using a First Ten Angstroms FTA 1000 C Class instrument, along with drop shape analysis software (DSA 100, Kruss, Hamburg, Germany). For static calculation of the contact angle, water droplets (2.0 μL) were applied on the surface every two seconds, and expansion of water was allowed. Regarding the reliability of contact angle measurement, the tangent-leaning method was employed for characterization of the contact angles on modified silicone implants [20]. The measurement was repeated 20 times for each sample.

Images of the surface roughness of the silicone implants were taken by an atomic force microscope (Nano Scope Multimode, Digital instrument, Bresso, Italy)). Prior to the AFM measurement, the samples were placed upwards onto a microscope slide and allowed to be dried in vacuo for 24 h. Surface imaging (3 × 3 µm dimension) was recorded in non-contact mode using a silicon tip on a nitride lever coated cantilever (125 µm length, PPP-NCHR 10M; Park Systems) under a resonance frequency of 200 to 400 kHz, a nominal force constant of 42 N/m, and a scan frequency of 1 Hz per line.

### 2.4. IL-4 Release Profiles

IL-4 release from the silicone breast implants was assessed by measuring the amounts of protein in the wash solutions. Initially, an IL-4-coated implant was placed in 1 mL of media at 37 °C with 5% CO_2_. At specific time intervals (1 h, 2 h, 4 h, 10 h, 1 day, 2 days, and 3 days), media were taken and replaced with fresh media. The amount of IL-4 in each sample was then determined using the ELISA method and calculated using a standard curve.

### 2.5. Macrophage Cell Culture for In Vitro Analysis

RAW 264.7 cells (ATCC, Rockville, MD, USA) were used in this study for the in vitro analysis. Cells were cultured in Dulbecco’s modified Eagle medium (DMEM) supplemented with 10% heat-inactivated fetal bovine serum (FBS), 100 U/mL penicillin, and 100 µg/mL streptomycin (Gibco, Carlsbad, CA, USA) in an incubator at 37 °C with 5% CO_2_. The RAW 264.7 cells were coated on 24-well plates at a cell density of 5 × 10^5^ cells/mL. Silicone implants immobilized with or without IL-4 were then placed on the cell culture plates. The silicone implants were capable of inducing biomaterial-mediated inflammation and activating macrophages, and lead to macrophage polarization [21,22].

### 2.6. Enzyme-Linked Immunosorbent Assay (ELISA)

The in vitro expression profiles of cytokine biomarkers from RAW 264.7 cell supernatants, including IL-6, TNF-α, IL-10, and IL-4, were assayed by ELISA to assess the macrophage polarization. The assay was performed by collecting supernatants at 24, 48, and 72 h from both IL-4-immobilized and non-immobilized samples. Each antibody in PBS was coated on 96-well plates for 24 h at RT. Then, PBS solution was used twice to clean the plates before blocking with 10% FBS in PBS for 2 h at RT. Later, cell supernatants were incubated in the plates, followed by incubation of biotin-conjugated secondary antibodies as well as streptavidin–horseradish peroxidase (HRP). A substrate of HRP, 2′-azino-bis (3-ethylbenzithiazoline-6-sulfonic acid), containing 30% hydrogen peroxide solution, was incubated, and the resultant UV absorbance was read in a microplate spectrophotometer at a wavelength of 405 nm (EPOCH2, BioTek, Winooski, VT, USA). The ELISA experiments were performed in duplicate.

### 2.7. Immunofluorescence Staining and MTT Assay

Fluorescence-activated cells were quantitatively assessed to measure the response of macrophage populations to IL-4. For the in vitro immunofluorescence staining assay, CD206 was used as a target marker to identify the M2 macrophages. Cells placed in the well plates were washed homogenously with PBS (pH 7.4) thrice for 5 min each before trypsinization. Afterward, a blocking solution (0.2% Triton X-100, 1% BSA in PBS) was applied in the plate for 1 h; later, the plates containing diluted primary antibodies (CD206 (Abcam, Cambridge, UK) were incubated overnight in a freezer at 4 °C. The next 1–2 h consisted of incubation steps at RT with the secondary antibodies—which were diluted to 1:2000—after washing the plates three times. The FITC used for defining CD206 was inserted and incubated for 30 min at 40 °C, followed by cleaning of the plates using PBS solution. Then, cells in the plates were stained with DAPI (VECTASHIELD, Vector Laboratories, Burlingame, CA, USA) after ensuring that they were washed thoroughly with PBS to stain cell nuclei. The cell polarization was checked, and then images of cells were taken under a confocal microscope with a z-stack. The ratio of M2/M1 macrophages was calculated by counting the absolute numbers of CD206-positive (M2) and CD206-negative (M1) cells. Cytotoxicity was determined using the MTT assay at 24, 48, and 72 h. Cells were incubated on both implant types for different time periods, followed by removal of the media, the addition of a 100 μL MTT (5 mg/mL) solution with fresh media, and incubation in a CO_2_ chamber. The solution was replaced by the addition of DMSO to solubilized MTT. Finally, the extracted solution was measured by an ELISA reader at 560 nm.

### 2.8. Reverse Transcription Polymerase Chain Reaction (RT-PCR)

RNA from cells was extracted and quantified using an easy-BLUE RNA extraction kit (iNtRON Biotechnology, Seongnam, Korea). The AccuPower^®^ RT PreMix (Bioneeer Corporation, Daejeon, Korea) was used to handle the extracted RNA, which was converted to cDNA. We designed and used a combination of two forward and two reversed primers, as shown in Table 1. The PCR settings were as follows: the annealing temperature was 62 °C for GAPDH and 60 °C for Arg-1. The presence of RNA amplicons was verified with a 2% agarose gel containing ethidium bromide under electrophoresis.

### 2.9. In Vivo Experiment

Nine-week-old Sprague–Dawley rats weighing 250–300 g were used for in vivo experiments. The rats weighed around 250–300 g and were randomly divided into 2 groups, including 5 rats per group. The animals were kept in a specific-pathogen-free (SPF) environment with fresh food and water, with a light/dark cycle of 12/12 h. The in vivo experiment followed the protocol of the Seoul National University Bundang Hospital Institutional Animal Care and Use Committee (approval number: BA1801-240/011-01), and the all methodological proceedings were carried out under the NIH Animal Care and Use Guide (NIH).

After anesthetizing the animals with isoflurane inhalation (Hana Pharm, Seoul, Korea), implant shells were inserted into the animals following the implant insertion procedure. Dorsal hair in the surgical area was shaved and disinfected with 70% alcohol and betadine. Afterwards, a laceration of 2–3 cm in length was incised on the dorsal region using a #15 scalpel blade, and the implant was placed at the sub-panniculus pocket. The incision region was stitched with a surgical suture (Nylon 4/0, Ethicon, Somerville, NJ, USA). The cascading events of inflammation were analyzed at proper intervals for tissue biopsy from each group. After the chosen animals were sacrificed, all components of the surgical sites, including the epidermis, dermis, and posterior and anterior capsules in the dorsal area, were selected for biopsy examination, and the implant was extracted. To fix the tissues, 10% paraformaldehyde was applied to all samples for 24 h at 4 °C, and they were embedded in paraffin.

### 2.10. In Vivo Evaluation of Capsule Thickness and Collagen Density

Firstly, a tissue block of paraffin was cut into slices, 4 μm thick. The chemicals xylene and ethanol were used for extraction from the diaphragm. The slides recorded in our previous study were checked for parameters such as macrophage polarization, capsule thickness, collagen density, and fibroblast and myofibroblast quantities. The capsule thickness was measured from H&E-stained tissue samples under 40× magnification on a microscope (LSM 700, Carl Zeiss, Oberkochen, Germany). The thickness was delineated from the covered area of the silicone insertion to the connected layer of the dorsal subcutaneous muscle. Three pictures were taken of three different parts of the capsule and used to determine the average thickness of the capsule with ZEN software (ZEN 2.3 blue edition, Oberkochen, Germany).

The collagen density was analyzed as previously described by Yoo et al. [23]. Briefly, ImageJ software was used to measure the blue-stained collagen area. The blue area was then divided by the percentage value of the total area to calculate the collagen density. Three sites were randomly chosen and analyzed. The macrophages, fibroblasts, and myofibroblasts were measured using the immunofluorescence (IF) staining method [23]. While CD11b and CD68 were used as pan-macrophage markers for mouse models, iNOS was used for only M1 macrophages, and CD206 and Arg-1 were used as specific markers for M2 macrophages. Rabbit anti-CD68 antibody and mouse anti-iNOS antibodies diluted to 1:300 were used to determine the M1 macrophage counts. Subsequently, rabbit anti-CD11b antibody and mouse anti-Arg-1 antibodies diluted to 1:300 were used to estimate the M2 macrophage counts. For myofibroblasts and fibroblasts, mouse anti-α-SMA antibody (diluted to 1:50) and rabbit anti-vimentin antibody (diluted to 1:250) were used, respectively. By using anti-rabbit secondary antibodies and anti-mouse antibodies (diluted to 1:2000), fluorescence signals were obtained. For all dilutions, 1× PBS containing 1% BSA and 0.1% SDS was used. Following the secondary antibodies, DAPI (H-1200; Vector Laboratories, Burlingame, CA, USA) in VECTASHIELD was used to mount the tissue slides. The cells (macrophage, fibroblast, and myofibroblast) associated with fibrosis were counted in an image area of 0.48 mm^2^ (200 × magnification), and images from each sample were randomly recorded at three other points in the capsule region.

### 2.11. Statistical Analysis

All the experimental results were analyzed using SPSS software (SPSS version 20; IBM SPSS, Armonk, NY, USA) and expressed as mean ± standard deviation. One-way ANOVA with Tukey’s post hoc test and Student’s *t*-test were used to compare the different treatment groups in vitro and in vivo, respectively. All *p*-values less than 0.05 (typically ≤0.05) denote statistical significance, which was used for all inferential statistics and the datasets.

## 3. Results

### 3.1. Characterization of the Modified Surface of Our Silicone Implants

A schematic representation of the preparation of an IL-4-immobilized silicone implant is exhibited in Figure 1. The surface morphologies of the modified silicone implants were investigated by AFM before coupling with IL-4. To monitor the effect of surface treatment on the silicone implant, the height distribution of the surface topography was found by reflecting the roughness of the modified layers. The surface root mean square roughness (Rq) values of intact, O_2_-plasma-treated, and APTMS-treated silicone implant shells were 2.06, 2.24, and 2.84 nm, respectively (Figure 2a–c). The Rq value of the implant treated with the bifunctional NHS linker (bis-dPEG^®^ _13_-NHS ester) was 10.4 nm (Figure 2d). When IL-4 was introduced, relatively low roughness of the height image (Rq = 6.14) resulted (Figure 2e). The presence of an amino group in the APTMS silicone surface was estimated by determining the nitrogen content. The energy dispersive X-ray spectroscopy (EDS) analysis revealed that the nitrogen content was 3.17% after treatment with APTMS (data not shown). The surface wettability of the silicone implant was determined by the static water contact angle measurement, as shown in Table 2. While strong hydrophobicity (water contact angle (WCA) value: 93.90°) was shown on the intact silicone implant surface, the WCA value after O_2_ plasma treatment was 0.16°. The WCA measurements highlight that APTMS-immobilized surfaces (WCA: 97.80°) can be slightly less hydrophilic than uncovered ones (WCA: 93.90°). In contrast, with IL-4 treatment, the WCA value decreased to 78.10°, and the bis-dPEG^®^ _13_-NHS ester-treated silicone implant layer with no IL-4 immobilization had a bigger value (100.60°), which proves that IL-4 was successfully introduced on the NHS ester-treated silicone implant layer. These results indicate that the IL-4 immobilization on the silicone implant surface could be monitored by surface wettability measurements.

### 3.2. In Vitro Release Profiles and the Effect of the IL-4-Coated Silicone Implant on Macrophage Polarization

The RAW 264.7 cells incubated with the silicone implant without IL-4 are denoted as “silicone,” and the cells incubated with the IL-4-immobilized shell are denoted as “IL-4-coated silicone” (Figure 3a). Figure 3b shows the cumulative release profile of IL-4 from the coated breast implant in culture media. Initially, the IL-4 release increased; however, over time, it showed sustained release from the implant. The total IL-4 released from the implant was approximately 7.99 ng/mL after 72 h. The release profile data revealed the strong bonding of IL-4 to the silicone, to which we attributed the decrease in inflammation in the adjacent area. Furthermore, the cell viability study showed that both implant types had suitable cellular behavior and did not elicit a toxic effect (Figure 3c). Analysis of Arg-1 as a potential marker for M2 macrophages was performed for both groups. PCR analysis of the cDNA revealed higher expression of the Arg-1 gene in the IL-4-coated silicone, and amplification of the GAPDH gene as a positive control showed similar amplicons with the same band intensity in both groups (Figure 3d). The fluorescent images revealed higher expression of CD206 as a biomarker for M2 in the IL-4-coated silicone when compared with the bare silicone (Figure 3d). The production of IL-6 and TNF-α as pro-inflammatory cytokines and IL-4 and IL-10 as anti-inflammatory cytokines was compared for both groups (Figure 3e,f). IL-6 is usually associated with M1 macrophages, but some recent studies have revealed IL-6 expression in M2 macrophages [24]. Although the pro-inflammatory activity of IL-6 is described in the literature, its capability to induce M2 macrophage polarization has recently been mentioned [25,26]. Based on recent studies, it was believed that IL-4-induced M2 macrophages only slightly expressed IL-6 [27]. Indeed, IL-4-coated implants showed similar IL-6 expression to uncoated silicone implants. However, TNF-α on the IL-4-coated silicone was downregulated significantly in comparison to bare silicone. Furthermore, the production of both IL-4 and IL-10 was upregulated over the course of 72 h (Figure 3f), both of which are crucial biomarkers for M2 macrophages.

### 3.3. Effect of the IL-4-Coated Silicone Implant on the Fibrous Capsule Formation In Vivo

The in vivo effect of the IL-4-coated silicone implants on capsular formation around the implants was investigated by histochemical and immunohistochemical analyses in a rat model (Figure 4a). Fibrous capsule development was determined by capsule thickness on the contact site of each implant. H&E-stained images were obtained to estimate the capsules’ thicknesses (Figure 4b). The tissue thickness of each capsule was determined, and at least three parts of each picture were analyzed per group. The wall diameter of the capsules around the bare silicone implants was significantly thicker than that around the IL-4-coated silicone implants (Figure 4b). The average capsular thickness was 629.4 μm around the bare silicone implants, and it was 317.6 μm around the IL-4-coated silicone implants—a statistically significant (*p* < 0.05) ca. 50% reduction in capsule formation caused by IL-4 coating (Figure 4c).

Furthermore, the collagen density in the tissues biopsied from all testing groups was evaluated by MT staining of the sliced sections [28]. Collagen density analysis was performed using the quantitative assessment function in ImageJ. Blue regions in the images were selectively extracted and their areas were calculated (Figure 4d). There was a significantly higher collagen density percentage on the bare silicone surfaces, an 80.14 ± 0.92% increase (Figure 4e), whereas a significant reduction in the rate of MT-positive tissue was observed on the IL-4-coated silicone surface (56.38 ± 8.33%) (Figure 4e), resulting in a 29.6% reduction in collagen density.

### 3.4. The Effect of the IL-4-Coated Silicone Implant on Macrophage Polarization

An immunofluorescence assay was performed to evaluate the effect of IL-4 on the polarization of macrophages after one week. Tissues were stained with M1/M2-specific markers and observed under a confocal microscope. To study whether the IL-4-coated silicone implants underwent a macrophage polarization switch, immunofluorescence analysis was performed for the expression of iNOS and Arg-I in M1 and M2, respectively. CD68 is a pan-macrophage marker usually used for detecting macrophages, and inducible nitric oxide synthase (iNOS) expression was related to M1 macrophages. Thus, CD68/iNOS were detected as the fraction of M1 macrophages in the total macrophages in capsular tissue [29]. A significant reduction in the number of CD68^+^ iNOS^+^ double positive macrophages (M1 macrophage) was observed in IL-4-coated silicone when compared to bare silicone (*p* < 0.05) (Figure 5a,b). On the other hand, a significant increase in double positive M2 cells, i.e., CD11b^+^ Arg1^+^ macrophages (M2 macrophage), was observed on the capsule lesions in IL-4-coated silicone (*p* < 0.05) when compared to bare silicone (Figure 5c,d).

### 3.5. Estimation of the Numbers of Fibroblasts and Myofibroblasts

Macrophages influence fibroblasts, which are known to play an important role in collagen synthesis and mature into myofibroblasts, which in turn lead to tension and induce CC formation [30]. Hence, the fibrosis of these extracellular tissues was displayed with immunofluorescence images captured by staining the cells with vimentin. The number of fibroblasts significantly increased to 122.8 ± 7.8 on the bare silicone (Figure 6a,b). On the IL-4-coated silicone, there were 32.4 ± 3.4 fibroblasts, or 73.6% fewer than on the bare silicone (*p* < 0.05, Figure 6). Additionally, myofibroblasts play an important role in triggering fibrosis; thus, in the current study, the number of myofibroblast cells around the silicone implants was estimated [31]. As shown in Figure 6c,d, on the bare silicone, the number of myofibroblasts significantly increased to over 91.6 ± 13.6 cells, whereas the number of myofibroblasts on the IL-4-coated silicone was 35.4 ± 2.7, or 61.3% fewer than on the bare silicone (*p* < 0.05, Figure 6c,d).

## 4. Discussion

In the current study, we have confirmed that IL-4-coated silicone implants promote M2 macrophage polarization and restrict inflammatory activities, which further inhibited fibrous capsule formation around the implants.

Implantation of silicone breast implants in the body generates a series of simultaneous reactions called the foreign body response, which facilitates the removal or isolation of the implants from the host tissues [32]. This response decreases the longevity and activity of the silicone breast implants. Moreover, serum proteins non-specifically adsorb throughout the implants, followed by activation of immune cells and coagulation cascades. The innate immune cells, such as neutrophils, dendritic cells, macrophages, natural killer cells, and T and B cells, are responsible for mediating an adaptive response [33,34,35]. Although the immune response is desirable for appropriate wound healing, chronic inflammation and releasing reactive oxygen intermediates can cause adverse biological events [36]. In addition, macrophage-frustrated phagocytosis causes the formation of foreign body giant cells and causes gathering fibroblasts to create fibrous tissue, which encapsulates the implants to confine them [32,36,37]. As a result, impaired wound healing occurs [38]. While different kinds of cells are involved in the foreign body response, macrophages are considered to be a crucial determinant of fibrotic capsular formation (Figure 1). To improve wound healing and tissue remodeling, macrophages can be polarized to a spectrum of phenotypes in response to their microenvironment [10,39]. Initially, immune cells release several types of cytokines and chemokines, including monocyte chemoattractant protein-1 (MCP-1), macrophage inflammatory protein-1 (MIP-1), IL-6, TNF-α, and IL-1β, in response to injury, and they support the recruitment of additional leukocytes and enhance their activation [40]. Immediately after implantation, pro-inflammatory cytokines such as IL-6, IL-1β, and MIP-1 dominate, and anti-inflammatory cytokines such as IL-10, IL-13, and IL-4 gradually increase [41]. Macrophages are highly sensitive cells that transform their phenotypes based on the cytokines present in the surrounding microenvironment [42,43]. Thus, local delivery of cytokines at the defect site for wound healing emerged as an innovative approach for rapidly resolving inflammation after biomaterial implantation (Figure 2c). In this study, we developed a silicone breast implant coated with IL-4, a Th2 cytokine inducing M2 cells. The surface modification of silicone was monitored using AFM, EDS, and WCA analyses. The changes in their values after each modification step showed that the surface was properly modified and coated with IL-4 (Figure 3). Our research verified that IL-4-coated silicone implants significantly induce a phenotypic transformation of macrophages from M0 to M2 in vitro, causing drastically reduced pro-inflammatory cytokine levels (TNF-α and IL-6) and increased anti-inflammatory cytokine levels (IL-4 and IL-10). These data suggest that IL-4-coated silicone implants might alleviate inflammation by inducing M2 macrophage polarization.

Having proved that IL-4-coated silicone implants can successfully alleviate inflammation by inducing M2 macrophage polarization, this study further determined the protective effect of the IL-4-coated silicone implants against fibrous capsule formation. Our in vivo study analyzed the tissue inflammation, macrophage phenotypes, density of collagen, capsule width, and fibroblast and myofibroblast counts, which are closely related to capsule contracture. Collagenous capsules are formed in response to the foreign body reaction after silicone breast implantation. Externally, a capsule develops as a relatively undetectable form of a thin membrane that increases breast size. However, a stronger foreign body response induces an excessive, hypocellular, thicker capsule that is rich in collagen, resulting in contracture formation [44]. Previous studies found that high M1/M2 concentrations could lead to the foreign body reaction, resulting in dense capsule formation [4]. M1 macrophages are considered to be involved in the synthesis of collagen, fibroblasts, and myofibroblasts, leading to a high prevalence of capsular contracture [1]. This has a great impact on neovascularization and dense fibrous capsule formation [45]. In addition, the pro-inflammatory cytokine IL-6, secreted from M1 macrophages, assists in the formation of fibrosis. Thus, limiting M1 macrophages is beneficial for reducing excess ECM production [46]. On the other hand, the role of M2 macrophages in fibrous formation depends on the damage site and the repairing behavior. If the damage is sustained for a long time, it facilitates epithelial and endothelial-to-mesenchymal transitions, and fibrocyte proliferation. [47]. However, the pivotal role of M2 macrophages lies in their release of anti-inflammatory cytokines that reduce inflammation and fibrous tissue formation. In this study, the impact of IL-4 on capsule thickness was evaluated. The results of H&E-stained images and the collagen density analysis support that capsule formation was reduced when using IL-4-coated silicone implants, in comparison to the regular implants (Figure 4). Furthermore, M1/M2 macrophage polarization was investigated at the capsule lesions. Unsurprisingly, IL-4-coated silicone implants promoted M2 macrophage polarization in fibrous capsules around the implants. In addition, immunohistochemical studies revealed that the populations of fibroblasts (visualized with vimentin staining) and myofibroblasts (visualized with α-SMA staining) were reduced in the IL-4-coated silicone implants (Figure 5). In the tissues around silicone implants, TGF-β secreted from fibroblasts facilitates collagen synthesis and promotes fibroblast differentiation into myofibroblasts [48,49]. These myofibroblasts generate an anti-contractile force, leading to fibrotic capsular contraction in addition to excessive deposition of collagen [50]. In such cases, myofibroblasts were found with contraction procedures, such as tenosynovitis, Dupuytren’s contracture, and fibrous implant capsule formation [51,52]. Therefore, these results suggest that the activation of M2 macrophages, induced by the IL-4-immobilized silicone implants, reduced inflammation and decreased capsule formation and collagen density. Though the current study has demonstrated the performance of IL-4-coated implants in terms of macrophage polarization and reducing capsule contracture, due to limited facilities and funding, more groups, including a control, could not be included. Therefore, it would be better to evaluate the polarization with other cell lines in future in vitro studies. Moreover, the capsule contracture could be better explained by inserting implants into the breasts of a larger animal, which will be considered in the future.

## 5. Conclusions

We propose that local delivery of IL-4 is an innovative strategy for alleviating capsule contracture and inflammation at implant sites. IL-4-coated silicone implants significantly reduced the collagen density and capsule thickness compared to silicone implants without IL-4 in a rat model. Notably, local IL-4 exposure decreased the M1 macrophage population, implying reduced pro-inflammatory cytokine secretion, whereas it increased the number of M2 macrophages, resulting in enhanced secretion of pro-healing cytokines. As a result, fibroblast recruitment was drastically hindered, leading to a decrease in the myofibroblast population at the capsular tissue site, which eventually reduced the capsule contracture. Therefore, the sustained release of IL-4 from silicone implants proved to be efficient at reducing capsule contracture around silicone implants by promoting M2 macrophages.

## Figures and Tables

**Figure 1 polymers-13-02630-f001:**
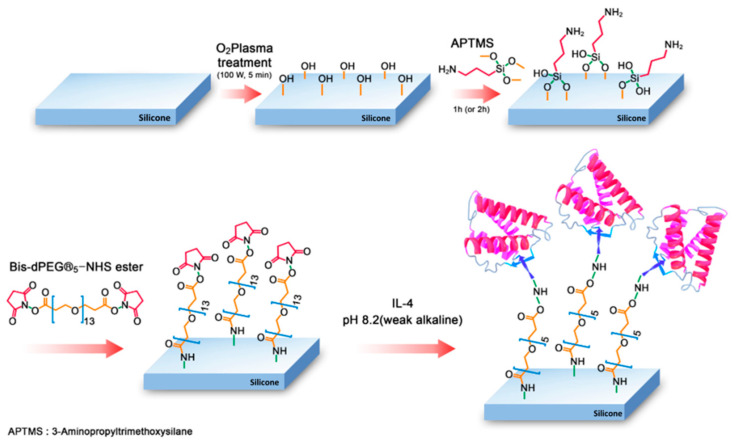
A schematic of IL-4 immobilization on a silicone implant. (i) O_2_ plasma activation of the silicone surface; (ii) silanization using APTMS; (iii) introduction of a bifunctional linker using a bis-dPEG^®^ 13-NHS ester; (iv) IL-4 immobilization under basic conditions.

**Figure 2 polymers-13-02630-f002:**
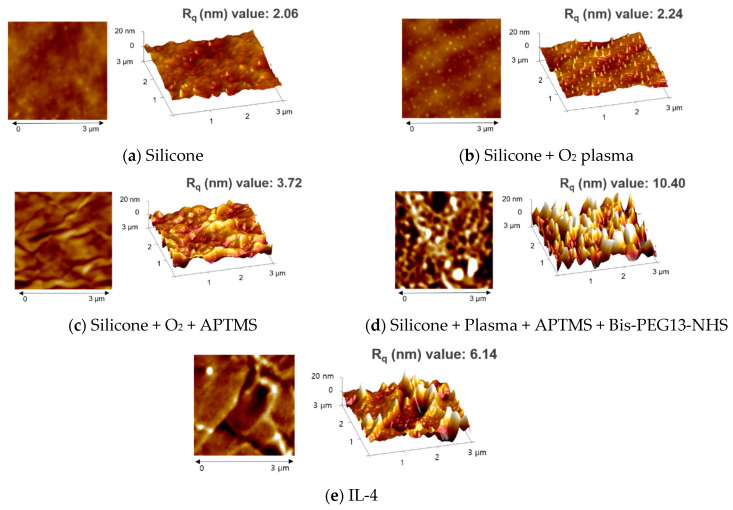
Surface morphologies of the silicone implants’ surfaces. The AFM images presented are surfaces of (**a**) a bare silicone implant, (**b**) O_2_ plasma-treated silicone implant, (**c**) APTMS and O_2_ plasma-treated silicone implant, (**d**) bis-dPEG^®^
_13_-NHS ester immobilization on (**c**), and (**e**) IL-4 immobilization on (**d**).

**Figure 3 polymers-13-02630-f003:**
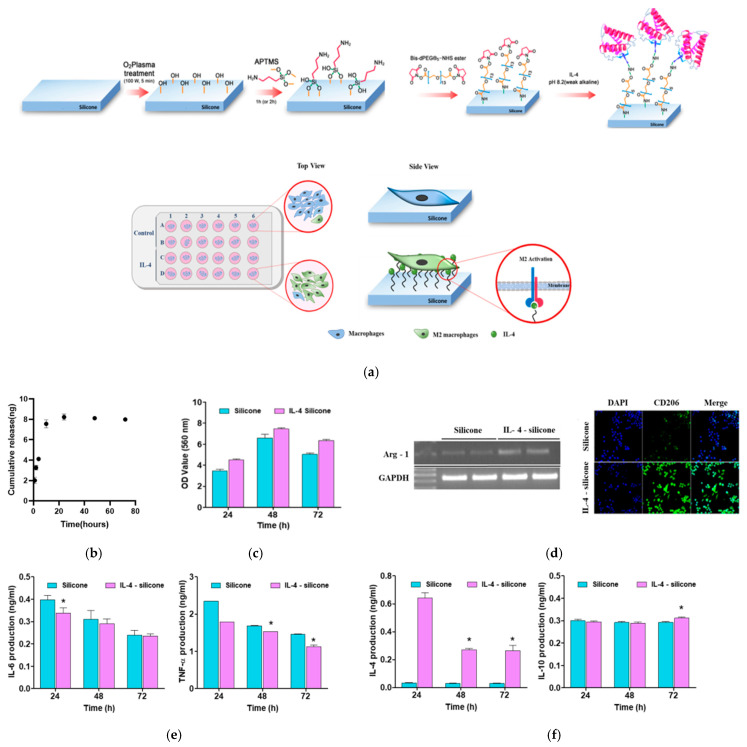
The effect of a silicone implant coated with IL-4 on macrophage polarization and the IL-4 release pattern in in vitro studies. (**a**) A schematic diagram of the in vitro cell culture system with IL-4-coated silicone implants. Raw 264.7 cells were cultured on the surfaces of silicone implants with or without IL-4 for 72 h. (**b**) IL-4 released from the silicone implant was evaluated at different time intervals. (**c**) MTT assay of silicone and IL-4-coated silicone. (**d**) Arg-1 expression on both implants and CD206 expression on the implants were assessed by confocal microscopy. (**e**) The production of pro-inflammatory cytokines (IL-6 and TNF-α). (**f**) Anti-inflammatory cytokines (IL-4 and IL-10) was determined by the ELISA method. The data are expressed as mean ± SD and show significantly (* *p* < 0.05; one-way ANOVA) different values from the bare silicone implant group.

**Figure 4 polymers-13-02630-f004:**
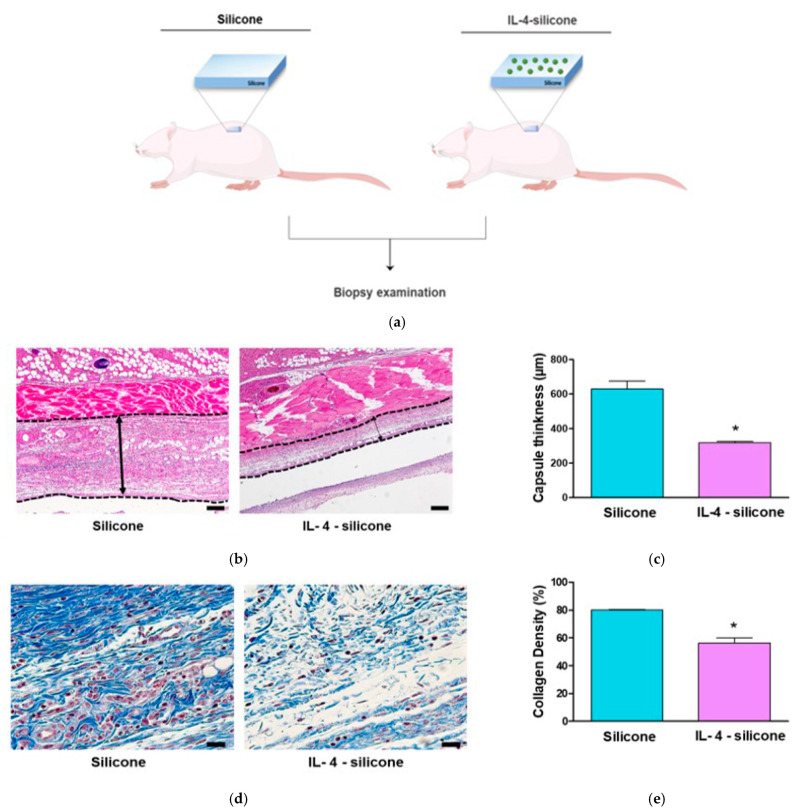
The effect of the IL-4-coated silicone implant on fibrous capsule formation in vivo. (**a**) A schematic diagram of the implantation of silicone implants with or without an IL-4 coating in SD rat. (**b**) Representative images of capsular formation around the silicone implants and (**c**) profiles of capsule thickness. The scale bars are 1 mm. The double-headed arrows indicate the capsule thickness. (**d**) Representative images of collagen formation around the silicone implants and (**e**) profiles of collagen density. The scale bars are 1 mm. The data are means ± SEMs, five animals per group. * Significantly different (*p* < 0.05; *t*-test) from the bare silicone implant group.

**Figure 5 polymers-13-02630-f005:**
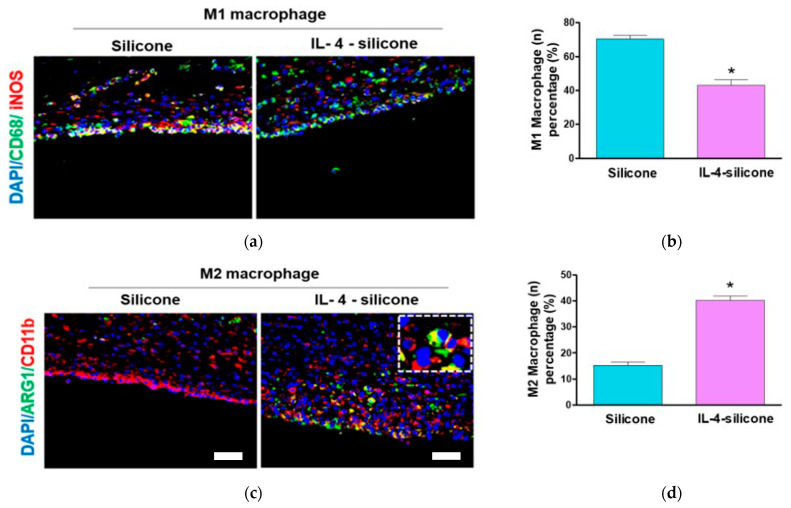
The effect of IL-4-coated silicone implants on macrophage polarization in vivo. (**a**) Representative CLSM images of tissues. Green fluorescence indicates CD68-positive macrophages. Red fluorescence indicates iNOS-positive macrophages. (**b**) Quantification of M1 macrophages. (**c**) Representative CLSM images of tissues. Green fluorescence indicates ARG-1-positive macrophages. Red fluorescence indicates CD11b-positive macrophages. (**d**) Quantification of M2 macrophages. Scale bars represent 50 μm. The data represent means ± SEMs, five animals per group. * Significantly (*p* < 0.05; *t*-test) different from the bare silicone implant group.

**Figure 6 polymers-13-02630-f006:**
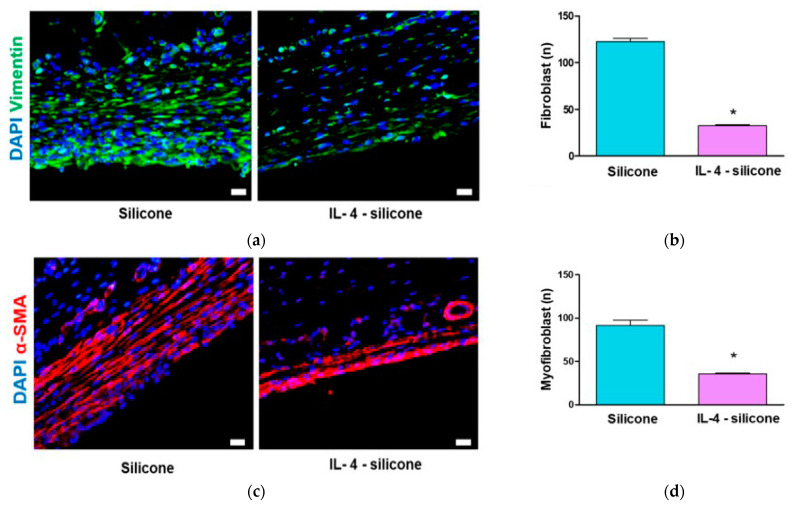
Evaluation of the numbers of fibroblasts and myofibroblasts around the implants. (**a**) Representative CLSM images of tissues. Green fluorescence indicates vimentin-positive fibroblasts. (**b**) Quantification of vimentin-positive fibroblasts. (**c**) Representative CLSM images of tissues. Red fluorescence indicates α-SMA-positive myofibroblasts. (**d**) Quantification of α-SMA-positive myofibroblasts. Scale bars represent 50 μm. The data represent means ± SEMs, five animals per group. * Significantly (*p* < 0.05; *t*-test) different from the bare silicone implant group.

**Table 1 polymers-13-02630-t001:** List of RT-PCR primers used in this study.

Gene-Specific Primers	RT-PCR Primer Sequence (5′-3′)
Arg-1 forward	5′-AAGAAAAGGCCGATTCACCT-3′
Arg-1 reverse	5′-CACCTCCTCT GCTGTCTTCC-3′
GAPDH forward	5′-GGC ATG GAC TGT GGT CAT GA-3′
GAPDH reverse	5′-TTC ACC ACC ATG GAG AAG GC-3′

**Table 2 polymers-13-02630-t002:** Water contact angle measurements of silicone breast implants.

Sample	WCA (°)
(a) Bare silicone prosthetic material (Si)	93.90
(b) Si/O2 plasma	0.16
(c) Si/O2 plasma/APTMS	97.80
(d) Si/O2 plasma/APTMS/Bis diPEG@13NHS ester	100.6
(e) IL-4 (cytokine immobilization)	78.1

## Data Availability

Not applicable.

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
