# Peer review of "Silicone Implants Immobilized with Interleukin-4 Promote the M2 Polarization of Macrophages and Inhibit the Formation of Fibrous Capsules"

_polymers, 2021, doi:10.3390/polym13162630_

Round 1
Reviewer 1 Report
Kim et al. have coated silicone implants with interleukin (IL)-4 to inhibit adverse immune reaction and dampen inflammation and the formation of fibrous capsules. The topic of this manuscript is interesting. However, I have some remarks and suggestions.
Major comments:
- The authors did not indicate which statistical tests were used for the analysis neither in Material & Methods nor in the figure legends.
- Moreover, in figure 3 a n-number of 3 is indicated. There is no statistical test allowing a comparison of such allow number. The authors should contact a bio statistician.
- The figures 3 (d) and (e) are not explained.
- In general, the image quality and resolution is low.
- Figure 3 (c) and (e): The decrease of cytokines over time is visible in both groups, silicone and IL-4 silicone. These effects could also be explained by dying cells. The authors should test for cell viability in these experiments, for example by analyzing lactate dehydrogenase amount in the supernatants in addition to cytokines.
Minor comments:
Line 41: it should be “300,378”
Line 91: what do the authors mean with stating “that is really honorable”?
Line 100: “was responsible”
Line 102: please define the abbreviation “FBS”
Line 260: Please state the used statistical tests.
Author Response
Comment 1: The authors did not indicate which statistical tests were used for the analysis neither in Material & Methods nor in the figure legends
Answer: Thank you for your comment. In this research, we evaluated the statistical value by t-test method. Since, all the experiments compared the two independent variable, t-test would appropriate. According to the author response, we have included the t-test in the 2.11 statistical analyses section and in the figure legends. This has been corrected in page 7 lines 283-284, page 10 line 357, page 11 line 384, page 12 line 405, and page 13 line 427.
Comment 2: Moreover, in figure 3 a n-number of 3 is indicated. There is no statistical test allowing a comparison of such allow number. The authors should contact a bio statistician.
Answer: Thank you for your response. In the figure3, we wanted to mention the experiment were done three times for the reproducible purposes. Mistakenly, we mentioned the comparison of three experimental data. According to your suggestion, we have corrected our mistake and clarify the respective part. This has been corrected in page 7 lines 283-284 and page 10 lines 351-358.
Comment 3: The figures 3 (d) and (e) are not explained.
Answer: Thank you for your response. According to your response, after modification the figure number we included the explanation. This has been added in page 10 lines 351-358.
Comment 4: In general, the image quality and resolution is low.
Answer: Thank you for your suggestion. According to your suggestion, we have improved the figure quality and resolution that embedded in the manuscripts.
Comment 5: Figure 3 (c) and (e): The decrease of cytokines over time is visible in both groups, silicone and IL-4 silicone. These effects could also be explained by dying cells. The authors should test for cell viability in these experiments, for example by analyzing lactate dehydrogenase amount in the supernatants in addition to cytokines
Answer: Thank you for your suggestion. According to your suggestion, we have provided the cell viability test by MTT assay. Our data showed, both implants were suitable cytocompatible and not significantly reduced the cells viability. This has been added in page 5 lines 211-215 and page 8 lines 332-333.
Minor comments:
Minor comments:
Line 41: it should be “300,378”
Answer: Thank you for your information. According to your comment, we have corrected the mistake. Changes of manuscript is written in red. This has been corrected in page 2 line 41.
Line 91: what do the authors mean with stating “that is really honorable”?
Answer: Thank you for the comments. According to the comment, we have changed the sentence and written in the manuscript. This has been corrected in page 3 lines 99-100.
Line 100: “was responsible”
Answer: Thank you for your information. According to your comment, we have corrected the mistake. Changes of manuscript is written in red. This has been corrected in page 3 lines 108.
Line 102: please define the abbreviation “FBS”
Answer: Thank you for your information. According to your comment, we abbreviate the FBS. Changes of manuscript is written in red. This has been corrected in page 3 lines 111.
Line 260: Please state the used statistical tests.
Answer: Thank you for your information. According to your comment, we have mentioned the used statistical test. Changes of manuscript is written in red. This has been corrected in page 7 lines 283-284.

Reviewer 2 Report
The article paper concerns the fabrication of novel IL-4 coated silicone implants. The results demonstrated that local IL-4 release transfers the M1 macrophages to M2 macrophages. This is a very interesting study. From my point of view, this work provides significant information regarding the ways to overcome the capsular contracture of silicone implants. However, the manuscript can be considered for publication after addressing the following concerns:
- As mentioned in the introduction, M2 macrophages are induced by Th2 cytokines such as IL-13, IL-10 and IL-4, and profoundly support tissue repair and growth. In your study, IL-4 was used for surface modification on silicone implant. Could you please explain why do not use the other Th2 cytokines?
- In the method, please explain how to sterilize the modified silicone implants before the implantation.
- The macrophages were induced by IL-4. Please use the macrophage subtypes of M2 macrophages. The following publication could be considered: DOI: 10.1016/j.actbio.2021.03.038
- In previous research, the addition of LPS (or PMA) into a cell culture often aims to create an inflammatory environment that activates the immune cells, skewing them towards a M1 phenotype. Please explain how to trigger immune cells to their pro-inflammatory response in your experiments (From M0 to M1).
- For the interpretation of results, please provide the data for the release kinetics of IL-4 in simulated body fluid.
- In Figure 3, could please explain that there were no obvious differences in IL-6 production at 48 h and 72 h.
- In previous study, M1 pro-inflammatory macrophages can be recognized by high expression
- of the cell surface marker CD86, while M2 anti-inflammatory macrophages can be recognized by CD206. In Figure 5, CD68 and iNOS were used for M1 macrophages. Please explain the reason. Also, please provide the scale bar in Figure 5 a and c.
- Please explain the limitation and outlook in the discussion.
Author Response
The article paper concerns the fabrication of novel IL-4 coated silicone implants. The results demonstrated that local IL-4 release transfers the M1 macrophages to M2 macrophages. This is a very interesting study. From my point of view, this work provides significant information regarding the ways to overcome the capsular contracture of silicone implants. However, the manuscript can be considered for publication after addressing the following concerns:
Comment 1: As mentioned in the introduction, M2 macrophages are induced by Th2 cytokines such as IL-13, IL-10 and IL-4, and profoundly support tissue repair and growth. In your study, IL-4 was used for surface modification on silicone implant. Could you please explain why do not use the other Th2 cytokines
Answer: Thank you for your invaluable suggestions and comments. We have modified the manuscript based on your comments and addressing your concerns.
IL-4, a multifunctional pleiotropic cytokine is expressed mainly by the activated T-cells also by eosinophils, basophils and mast cells. It maintain the typical cytokine structure and shares the sequence homology, cell surface receptors and intracellular signaling. M2 macrophages are also induced by other cytokines IL-10 and IL-13. However, IL-4 is necessary for CD4+ Th2 cell development, while IL-13 has been reported to have negligible effects on T cells. In addition some studies reporter IL-4 is more effective than IL-13. On the other hand some studies showed that IL-4 drastically enhanced the IL-10 production. It has been demonstrated, that Il-4 is best known for defining the Th2 phenotype and for maintain apoptosis, cell proliferation and expression of myriad genes in numerous cell types, including fibroblasts, macrophages, lymphocytes, and endothelial cells. Based on your comment, we further modify the introduction section mentioning the reason for choosing IL-4.
This has been corrected in page 3 lines 77-87.
Comment 2. In the method, please explain how to sterilize the modified silicone implants before the implantation.
Answer: Thank you for your comment. According to your response, we include the sterilization process in the 2.2 Immunobization of IL-4 on silicone implant section. This has been added in page 4 line 4 127 and 140.
Comment 3. The macrophages were induced by IL-4. Please use the macrophage subtypes of M2 macrophages. The following publication could be considered: DOI: 10.1016/j.actbio.2021.03.038
Answer: Thank you for your response. Based on your suggestion and the publication, we use the macrophage subtypes of M2 and include the publication in the references.
Comment 4: In previous research, the addition of LPS (or PMA) into a cell culture often aims to create an inflammatory environment that activates the immune cells, skewing them towards a M1 phenotype. Please explain how to trigger immune cells to their pro-inflammatory response in your experiments (From M0 to M1).
Answer: Thank you for your comment. In the current study, we have focused on macrophage polarization for decreasing the inflammation and reducing the capsule contracture. We were evaluated the impact of silicone implant on RAW 264.7 cells without inducing the inflammatory environment by using LPS (or PMA). However, we seeded cells on the silicone implant which induced biomaterial mediated inflammation. According to the previous studies,
Silicone implants have the capability to activate macrophages and led to polarization. Based on your suggestion, we included the macrophage polarization with references in section 2.5 Macrophage cell culture for in vitro analysis. This has been added in page 5 lines 178-180.
Comments 5: For the interpretation of results, please provide the data for the release kinetics of IL-4 in simulated body fluid.
Answer: Thank you for your response. According to your suggestion we have added the release profile data in the manuscript. This has been added in page 4 lines 164-170, page 8 lines 327-332, page 10 lines 353-354.
Comments 3. In Figure 3, could please explain that there were no obvious differences in IL-6 production at 48 h and 72 h.
Answer: Thank for your response.IL-6 is usually associated to M1, but some recent studies have revealed IL-6 expression in M2 macrophages. Though in literature the proinflammatory activity of IL-6 is described, its capability to induce M2 macrophage polarization has been mentioned. Based on the recent studies, we interpreted IL-4 induced M2 macrophages slightly expressed the IL-6. According to your comments, we have described our explanation with references in the manuscript. This has been added in page 9 lines 341-349.
Comments 4: In previous study, M1 pro-inflammatory macrophages can be recognized by high expression of the cell surface marker CD86, while M2 anti-inflammatory macrophages can be recognized by CD206. In Figure 5, CD68 and iNOS were used for M1 macrophages. Please explain the reason. Also, please provide the scale bar in Figure 5 a and c.
Answer: Thank you for your comments. In our current study, we used CD68/iNOS for detection of M1 macrophages in the rat tissues. Since CD68 is pan marker usually used to detect all macrophages, inducible nitric oxide synthase (iNOS) expression was related to M1 macrophage. Thus, CD68/iNOS were detected as the fraction of M1 in total macrophages at capsular tissue. Based on your comments, we have include reference with modification in the manuscript regarding the expression of CD68/iNOS in macrophage polarization. This has been added in page 11 lines 390-393.
Comments 5. Please explain the limitation and outlook in the discussion.
Answer: Thank you for response. According to your suggestion, we are explaining the limitation and outlook of this study in the discussion section. This has been added in page 14 lines 504-510.

Reviewer 3 Report
The manuscript entitled "Silicone implants immobilized with interleukin-4 promotes the M2 polarization of macrophages and inhibits the formation of fibrous capsules" authored by Kim et al. demonstrated the local delivery of IL-4 a novel strategy to manage the capsule contracture and inflammation associated with the silicone based breast implants. The study was well-formulated, technically sound and neatly presented. The manuscript may be considered for publication in Polymers journal.
Author Response
Thank you for your suggestion.
Round 2
Reviewer 1 Report
The authors have stated “The variability of data from three separate experiments were expressed as mean ± standard deviation. Student t-tests were conducted by SPSS software (SPSS version 20; IBM SPSS, Armonk, NY, USA).”. The n-number of three is not sufficient for any statistical analysis (all data in Figure 3). In the animal experiments (Figure 4 – 6) five animals were used per group and again, the t-test was used for statistical analysis. The t-test is a parametric test, but the n-number is to low to test for normal distribution. For this analysis, a non-parametrical test is needed (e.g. Mann-Whitney-U-Test). The authors must contact a bio statistician.
Author Response
The authors have stated “The variability of data from three separate experiments were expressed as mean ± standard deviation. Student t-tests were conducted by SPSS software (SPSS version 20; IBM SPSS, Armonk, NY, USA).”. The n-number of three is not sufficient for any statistical analysis (all data in Figure 3). In the animal experiments (Figure 4 – 6) five animals were used per group and again, the t-test was used for statistical analysis. The t-test is a parametric test, but the n-number is to low to test for normal distribution. For this analysis, a non-parametrical test is needed (e.g. Mann-Whitney-U-Test). The authors must contact a bio statistician.
Answer: Thank you for your suggestion. In our in-vitro as well as in-vivo studies, we used n-number of five samples in each experiment based on the literature reviews. In-vitro study was conducted triplet for evaluated the uniformity of results. In the research article we used One-way ANOVA with a Tukey post hoc and student t-tests to contrast different treatment group for in-vitro and in-vivo, respectively.
Reviewer 2 Report
The manuscript can be considered for publication
Author Response
Thank you for your suggestion.